# The Effect of Process Parameters on the Properties and Microstructure of A380 Aluminum Alloy Casting with Different Wall Thicknesses

He Li [1], Han Zhang [1], Wenfei Peng [1,*], Bo Lin [2,*], Yiyu Shao [1], Longfei Lin [1], Bangjie Fu [1] and Ziming Yu [1]

[1]   College of Mechanical Engineering and Mechanics, Ningbo University, Ningbo 315211, China; lihe@nbu.edu.cn (H.L.)
[2]   School of Mechanical Engineering, Gui Zhou University, Guiyang 550025, China
*   Correspondence: pengwenfei@nbu.edu.cn (W.P.); linbo1234@126.com (B.L.)

**Abstract:** In the present work, the effects of different die-casting process parameters on the mechanical properties and microstructure of A380 aluminum alloy casting with different wall thicknesses during the solidification process have been experimentally investigated. The experimental results show that both boost pressure and injection speed have a significant effect on the mechanical properties of the casting. As the injection speed increases, the changes in mechanical properties are more significant in the thin-walled area, while increasing the boosting pressure has a greater effect on the mechanical properties of the thick-walled area. In addition, the evolution of microstructure composition, including the $\alpha$-Al phase, eutectic Si phase and Al-Si-Fe-Mn phase, has been analyzed and compared by energy-dispersive spectroscopy (EDS), optical microscopy (OM) and scanning electron microscopy (SEM). It was found that the $\alpha$-Al phase in the thin-walled area is significantly refined with the increase of injection speed. Meanwhile, with the increase of boost pressure, the $\alpha$-Al phase in the thick-walled area gradually becomes finer, and the distribution of the eutectic Si phase and the Al-Si-Fe-Mn phase in the alloy becomes more uniform. Thus, the injection speed and boost pressure have an important impact on the overall forming quality of the casting.

**Keywords:** high-pressure die-casting; process parameter; A380 aluminum alloy; microstructure; mechanical properties



## 1. Introduction

The differential housing is a critical casting of the transmission system in sport utility vehicles. Due to the vehicle's high power output and starting torque, it is subjected to significant loads during operation and complex service conditions. Therefore, it requires good plasticity and toughness, as well as the ability to withstand a certain degree of plastic deformation. It must also have good sealing, weldability, and machinability, among other requirements [1]. Traditionally, the differential housing was manufactured through gravity casting, which resulted in high weight, low material utilization, and low dimensional accuracy, not meeting the demand for lightweighting of automobiles. Moreover, high-pressure die-casting is a process of injecting liquid or semi-solid metal into the cavity of a die-casting mold at high speed and pressure, allowing it to solidify and form under high pressure. This process has several advantages, including low cycle time, high cooling rate, good surface quality, the ability to cast complex near-net shapes with tight tolerances, and the capability to produce thin-walled structures [2]. In order to meet the demands of automotive lightweighting and obtain high-performance castings, A380 aluminum alloy material was selected instead of ductile iron, and high-pressure die-casting was used to form the differential housing.

In addition, the high-pressure die-casting process can produce products with complex shapes. However, due to limitations in mold structure, existing processes, and casting

methods, there are many unavoidable defects in the high-pressure die-casting process of aluminum alloy, making it a hot topic of die-casting research. Dargusch et al. [3] inserted a pressure sensor into the cavity of the die-casting mold to detect pressure changes during solidification and studied the effect of casting pressure on porosity defects in high-pressure die-casting. They found that increasing casting pressure can reduce porosity rates and that porosity rates increase with increasing casting speed. Verran et al. [4] used a combination of simulation and experimentation to study the effect of pressure injection parameters on the formation of $AlSi_{13}Cu$ alloy die-casting defects. The study showed that slow speed and high pressure conditions are conducive to producing better quality products and obtaining reasonable die-casting process parameters. Sun et al. [5] studied the effect of thin-walled magnesium alloy high-pressure die-casting molding porosity defects on its ductility. Gunasegaram et al. [6] demonstrated the effect of optimizing the pouring system and head-filling speed on the elongation of HPDC aluminum alloy. Increasing the speed of the pouring system and improving the flow channel structure can reduce porosity and improve the mechanical properties of A380 aluminum alloy. While most research centers on simulating die-casting process parameters and analyzing their effect on casting properties through experiments, the microscopic mechanism of their effect on properties is not clear. Therefore, it is crucial to understand the influence mechanism of different die-casting process parameters on casting microstructure and mechanical properties.

The current research on microstructure mainly focuses on small-scale simulations and experimental studies on a pattern scale. Timelli et al. [7] studied the significant microstructure characteristics of the high-pressure die-casting (HPDC) $AlSi_9Cu_3$ (Fe) alloy. They found that the die-casting microstructure consists of various microheterogeneous structures, such as positive eutectic segregation bands, primary Fe-rich intermetallics (sludge), and porosities. Otarawanna et al. [8] cast two special aluminum alloys, $AlSi_4MgMn$ and $AlMg_5Si_2Mn$, into tensile specimen bars using high-pressure die-casting. They characterized the microstructure of the tensile bar specimens and found that external solidification crystals (ESC), defect bands, surface layers, grain size distribution, and porosity were similar in both alloys. Wang et al. [9] analyzed the temperature field and flow field characteristics of ferritic stainless steel Cr17 during solidification in the mold using finite element and finite difference methods and found that low degrees of undercooling can prevent solidified shells from forming rapidly on the surface of the nucleus generator. Liu et al. [10] investigated the formation of grain structure and macroscopic segregation during the solidification of Al-4wt% Cu alloy using a multiscale fully coupled CAFE model. Gu et al. [11] developed a three-dimensional (3-D) model based on cellular automaton (CA) and process simulation to predict grain size of components produced by high pressure die casting (HPDC) of aluminum alloys. Yuan et al. [12] investigated the tissue evolution of die-cast A380 alloy during heat treatment. Through detailed analysis of macroscopic segregation, microstructure morphology, and grain size, they found that the different aging behavior exhibited in the surface and central regions was mainly attributed to the enrichment of Si and Cu in the surface layer and the presence of solidified crystals in the central region. Li et al. [13] investigated the effects of strontium (Sr), iron (Fe) and manganese (Mn) additions in the casting process on the microstructures and mechanical properties of $AlSi_7Cu_3$ alloy and found that addition of Sr slightly refines the eutectic silicon particles but that it also introduces more pores, while the volume fraction of porosity decreases with the Fe and Mn content increase. However, due to the spatial scale effect and non-uniform characteristics of the casting environment, stress concentration and severe non-uniform distribution of tissue composition may occur in local locations of the castings [14,15], which is especially evident in high-pressure die-casting. Hou et al. [16] studied the characteristics of defect bands in the microstructure of high-pressure die-cast AE44 magnesium alloy and found that process parameters and casting structure greatly influence the distribution of internal defect bands by affecting the content and degree of aggregation of solidification crystals (ESCs) outside the cross-section of the die casting. Song et al. [17] studied the aging behavior of high-vacuum die-cast Mg-9Al-1Zn magnesium alloy and found that the

microstructure of the surface of the cast specimen after aging was different from that of the center. Therefore, small-scale simulations and experimental studies of microstructure at the specimen scale cannot fully characterize the state of complex castings after solidification. Systematic studies of the microstructure of the castings as a whole after solidification are needed. Jiang et al. [18] successfully prepared high-quality WE43 magnesium alloy ingots with a diameter of 500 mm by direct chilling (DC) casting and found that the average grain size of WE43 magnesium alloy ingots gradually decreased from the center to the edge of the ingot. Lee et al. [19] investigated the phenomenon of inverse surface macro-segregation in high-pressure die-cast AM60 magnesium alloy. They found that local shrinkage occurs during solidification, when solidification starts at the casting surface, and provided metallographic evidence for the presence of reverse macroscopic segregation in HPDC AM60 magnesium alloy. However, there is still a lack of systematic studies on the microstructure of different wall thickness areas after the overall solidification of the casting with different die-casting process parameters.

In this study, an experimental investigation of high-pressure die casting of A380 aluminum alloy differential housing was carried out to examine the effect of different die-casting process parameters, namely injection speed and boost pressure, on the microstructure of castings with different wall thicknesses. Direct samples were taken from different wall thicknesses of the castings for microstructure observation, tensile property determination, and fracture morphology analysis. The research focused on different wall thickness areas of the castings to obtain the influence of die-casting process parameters on the organization and properties of these areas. The aim of this study was to provide theoretical guidance for actual production and manufacturing.

## 2. Experimental Procedures

### 2.1. Material and Tensile Tests

The material used for casting is A380 aluminum alloy, and Table 1 shows the main chemical composition of the material; the chemical composition of each casting is basically the same. As the bracket and bolster-hole connections bear most of the load in actual working conditions, the sampling positions of the tensile specimens under different die-casting process parameters (injection speed and boost pressure) are illustrated in Figure 1b. The MTS810 material testing tensile machine shown in Figure 1a is used to take three tensile specimens from each white part of the casting by a wire-cutting mechanism, and their mechanical properties are tested separately. The final result for the mechanical properties is the average of the three. Tensile tests were carried out using a MTS810 electronic testing machine at a strain rate of 2 mm/min at room temperature. Unidirectional tensile specimens were prepared according to the requirements of the national standard GB/T 16865-2013. The geometric dimensions of the specimen are shown in Figure 1c.

**Table 1.** The chemical composition of A380 aluminum alloys (mass fraction, %).

| Element | Si | Fe | Cu | Mn | Mg | Zn | Cr | Ti | Al |
|---|---|---|---|---|---|---|---|---|---|
| Concentration | 7.5~9.5 | 0.79 | 3.0~4.0 | 0.18 | 0.05 | 1.44 | 0.03 | 0.03 | Bal. |

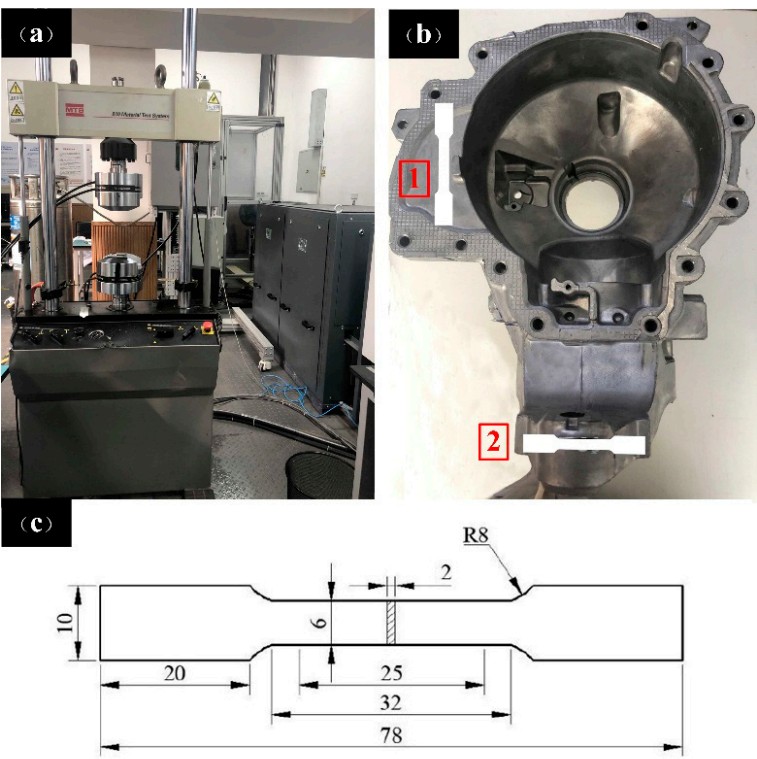

**Figure 1.** (**a**) MTS810 material testing system, (**b**) Sampling position of tensile specimen, (1) thin-walled area, (2) thick-walled area, (**c**) Size of uniaxial tensile.

### 2.2. Microstructure Characterization

To quantify the microstructural features, a combination of SEM, EDS, OM and image processing software was used. Metallographic samples were taken from the thin wall (position 1) and the thick wall (position 2) of the upper plate of the differential housing using a wire cutter—the actual thicknesses at the thick-walled area and thin-walled area are 18.8 mm and 6.2 mm, respectively—and then polished with 150#, 800#, 2000#, and 3000# sandpaper in turn. After inlaying, the specimens were cleaned with alcohol and dried with a hair dryer. EDS was used to analyze the microstructural components of different wall thickness areas, and microstructural comparisons were conducted under an optical microscope and a SU5000 scanning electron microscope.

## 3. Experimental Results

### 3.1. Mechanical Properties of Castings with Different Wall Thickness Areas

Figures 2 and 3 present the mechanical properties of tensile specimens tested under different injection speeds and boost pressures. The yield strength (Ys) and tensile strength (Ts) in the thick-walled area are slightly lower at an injection speed of 5.0 m/s. At an injection speed of 4.5 m/s, the Ys and Ts in the thick-walled area reach their maximum values of 141.1 MPa and 263.2 MPa, respectively, which are about 3.68% and 2.53% higher than those at 3.5 m/s. The Ys and Ts in the thin-walled area reached their maximum values of 136.2 MPa and 259.5 MPa, respectively, at an injection speed of 5.0 m/s, which are higher than those at 3.5 m/s. The Ys and Ts in the thin-walled area reach the maximum at 5.0 m/s with 136.2 MPa and 259.5 MPa, respectively, which are about 2.78% and 3.19% higher than those at 3.5 m/s. Regarding the elongation, the elongation in the thin-walled area is much larger than that in the thick-walled area, and the elongation in the thin-walled area increases with the increase of the injection speed, reaching a maximum value of 3.5% at the injection speed of 5.0 m/s, which is about 11.11% higher than that at 3.5 m/s. In contrast, the elongation in the thick-walled area slightly increases with the increase of the injection speed and then decreases significantly; the elongation at 5.0 m/s decreases by

about 7.38% compared to that at 3.5 m/s. Additionally, it can be observed from Figure 3 that the Ys and Ts of both areas increase with the boost pressure, but the thick-walled area experiences a more significant increase. Although the Ys and Ts are initially lower than the thin-walled area at 700 bar, with an increase in boost pressure the Ys and Ts of the thick-walled area gradually surpass those of the thin-walled area, reaching their maximum values at 850 bar, which are 141.1 MPa and 263.2 MPa, respectively. This represents an increase of approximately 34.28% and 30.19%, respectively, compared to 700 bar, while the thin-walled area only increases by about 12.19% and 21.03%. Regarding elongation, both areas experience an increase with the boost pressure, but the thin-walled area still exhibits a significantly higher elongation. At 850 bar, the elongation of the thin-walled and thick-walled areas reach their maximum values, which are 3.5% and 2.95%, respectively. These values are approximately 18.64% and 13.46% higher than those at 700 bar, respectively.

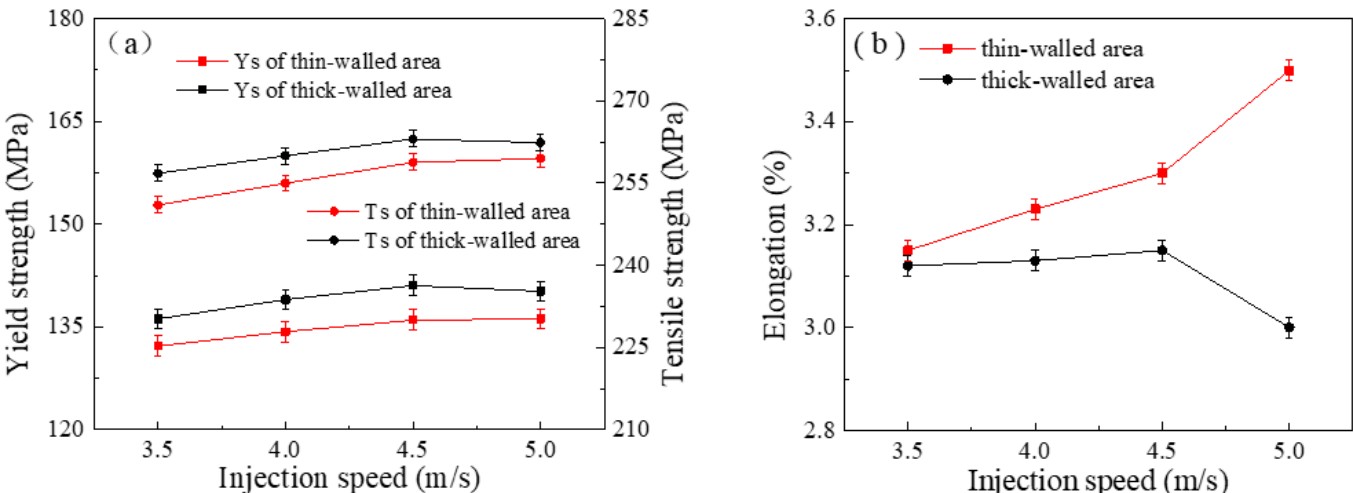

**Figure 2.** Mechanical properties: (**a**) Ys and Ts, (**b**) elongation of castings with different wall thicknesses under different injection speeds.

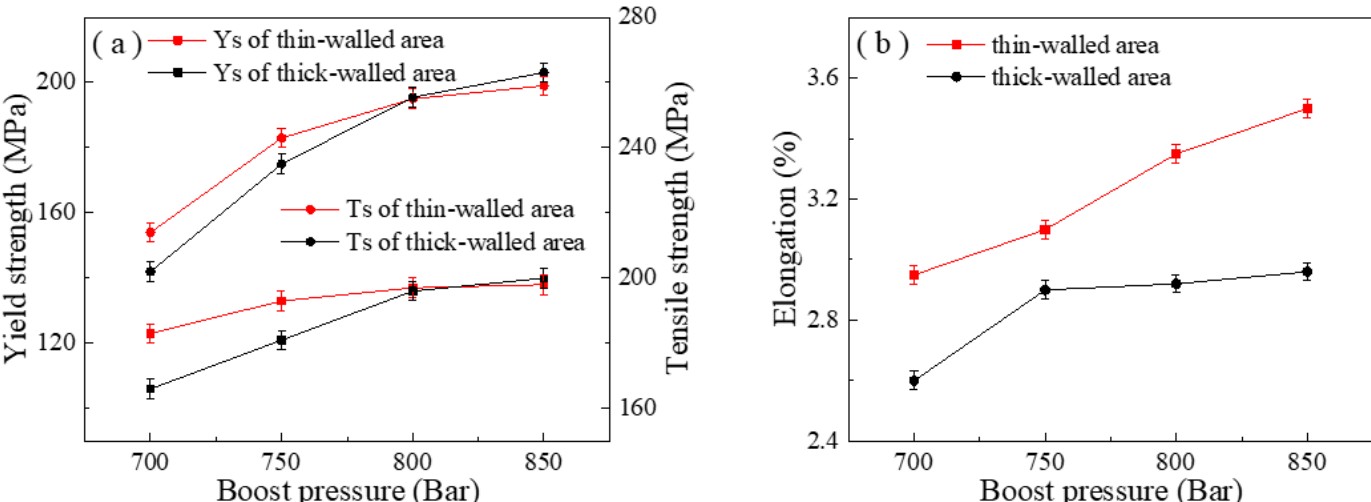

**Figure 3.** Mechanical properties: (**a**) Ys and Ts, (**b**) elongation of castings with different wall thicknesses under different boost pressures.

### 3.2. Microstructure Analysis

### 3.2.1. Microstructure Analysis at Different Wall Thicknesses

Figure 4 shows the microstructure of different areas magnified by 300 times under the casting process parameters of pouring temperature of 660 °C, mold preheating temperature of 200 °C, injection speed of 5.0 m/s and boost pressure of 850 bar. It can be observed

from Figure 4 that the microstructure and grain size distribution of the alloy are relatively homogeneous, comprised mainly of the primary α-Al phase, secondary α-Al phase, eutectic Si phase, Al$_2$Cu phase, and a small amount of the Al-Si-Fe-Mn phase. The primary α-Al phase exhibits dendritic distribution with low volume fraction, while the secondary α-Al phase and eutectic Si phase are the dominant phases, with the former exhibiting a dark gray spherical distribution. The eutectic Si phase exhibits a light gray distribution with different morphologies in different areas. The Al$_2$Cu phase is distributed at the grain boundaries and appears as a bright coral shape, while the Al-Si-Fe-Mn phase shows a polygonal distribution throughout the microstructure [20], as demonstrated in the EDS plot of Figure 5.

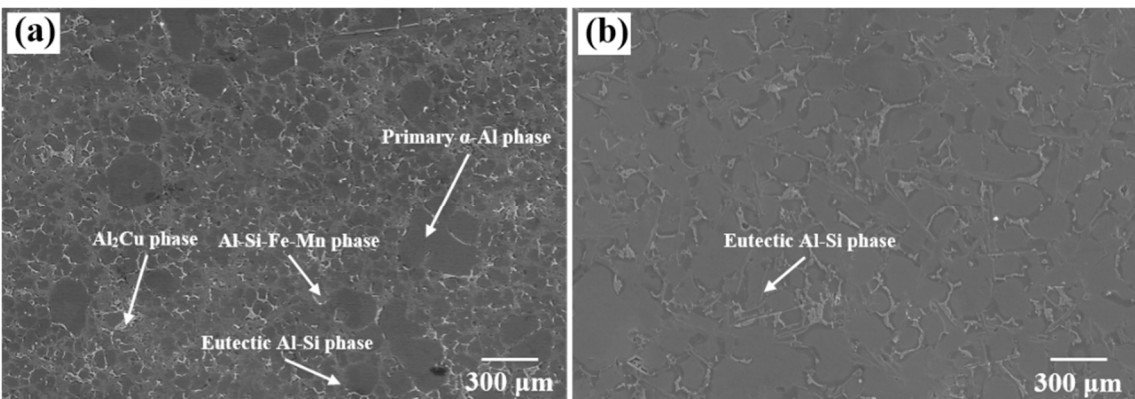

**Figure 4.** Microstructure of different areas of the casting: (**a**) thin-walled area, (**b**) thick-walled area.

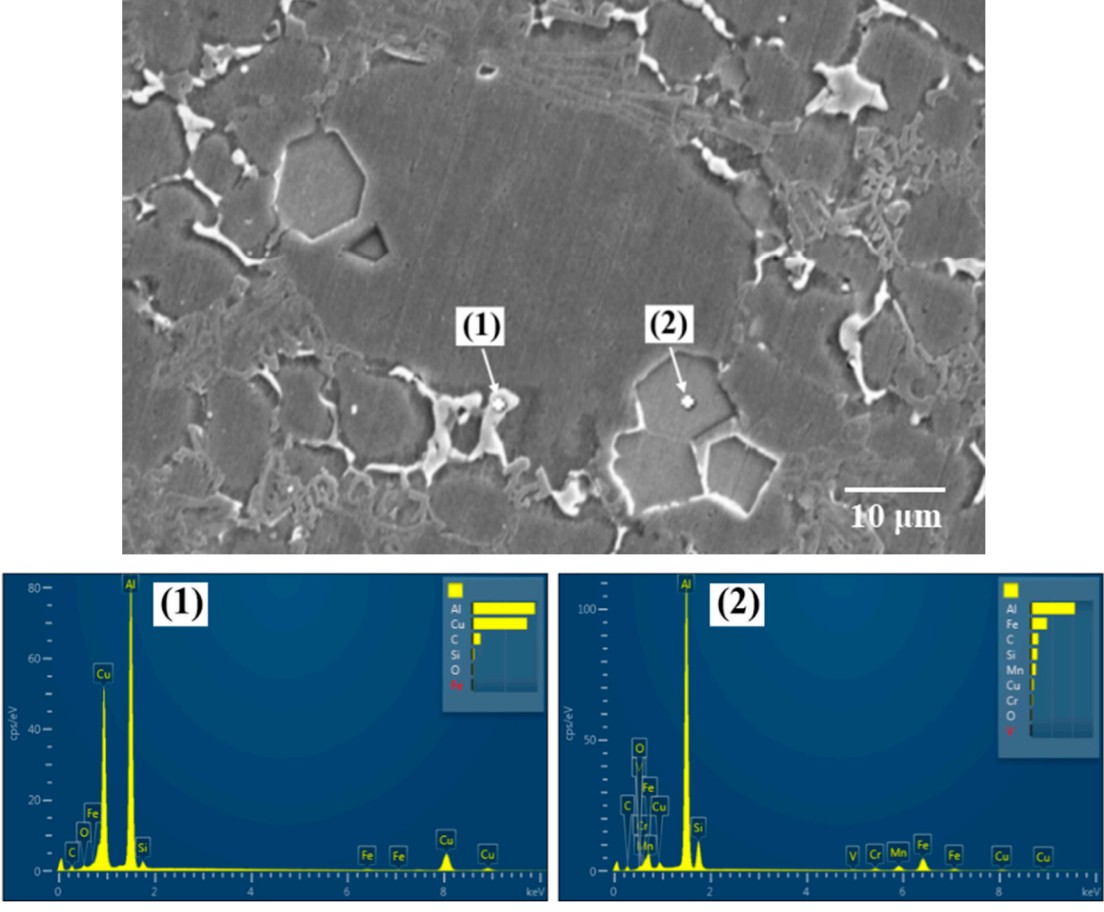

**Figure 5.** EDS images of Al$_2$Cu phase (1) and Al-Si-Fe-Mn phase (2).

Moreover, the microstructure of the two areas reveals that the matrix in the thin-walled area is finer and more evenly distributed than that in the thick-walled area, as shown in Figure 4. The uneven distribution of the $Al_2Cu$ phase in the thick-walled area is also more prominent than that in the thin-walled area. This can be attributed to the relatively fast heat transfer in the thin-walled area, resulting in large undercooling, which promotes the formation of fine grains and their uniform distribution. This process inhibits the precipitation and growth of the $Al_2Cu$ phase, resulting in smaller particle sizes compared to those in the thick-walled area, and positively impacting the mechanical properties of the thin-walled area. However, the $Al_2Cu$ phase in the thin-walled area is completely bonded to the grains, which can greatly reduce the mechanical properties of the area. On the other hand, in the thick-walled area, the $Al_2Cu$ phase becomes dispersed around the grains. Moreover, the amount of the Al-Si-Fe-Mn phase is higher in the thin-walled area and lower in the thick-walled area, which is also due to the larger subcooling in the thin-walled area compared to that in the thick-walled area.

In contrast, the thin-walled area of the casting exhibits mainly a short rod-like or dotted eutectic Si phase, as shown in Figure 6a, while the thick-walled area displays mainly a coarse needle-like eutectic Si phase, as shown in Figure 6b. This difference is attributed to the large subcooling in the thin-walled area, which increases the number of nucleated cores and improves the nucleation rate, resulting in the refinement of eutectic Si [21,22]. Although the short rod-like eutectic Si phase is less prone to cracking under external forces than the coarse needle-like eutectic Si phase, the finer eutectic Si and Al-Si-Fe-Mn phases tend to aggregate and combine, making the casting more susceptible to cracking, thus reducing the mechanical properties of the thin-walled area.

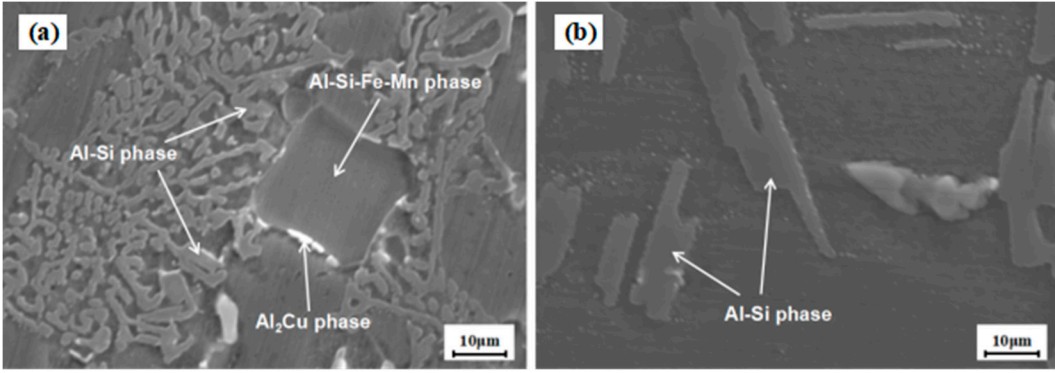

**Figure 6.** SEM images of different areas of the casting: (**a**) thin-walled area, (**b**) thick-walled area.

3.2.2. Effect of Injection Speed on the Microstructure at Different Wall Thicknesses of the Casting

Figure 7 shows the microstructure of the thin-walled area at different injection rates under the conditions of casting temperature of 660 °C, mold preheating temperature of 200 °C, and boost pressure of 850 bar. It can be seen that the average size of grains, eutectic Si phase and shrinkage pores in the thin-walled area gradually decrease with increasing injection speed. The average grain size in the thin-walled area is 19.76 μm, 19.03 μm, 18.56 μm, and 18.13 μm, the average length of eutectic Si phase is 9.76 μm, 9.68 μm, 9.54 μm and 9.43 μm, and the average diameter of shrinkage pores is 7.68 μm, 6.55 μm, 6.56 μm and 5.72 μm when the injection speed is 3.5 m/s, 4.0 m/s, 4.5 m/s and 5.0 m/s, respectively, as calculated by the IPP software. At an injection speed of 3.5 m/s, as shown in Figure 7a, the microstructure displayed non-uniformly shaped primary α-Al phases with larger sizes and a large number of smaller spherical secondary α-Al phases. Additionally, pores and noticeable eutectic Si segregation regions can be observed in the tissue, as indicated by the black areas in the metallographic figure. This can be attributed to Si being the leading phase in the eutectic reaction, while Al and Si belong to the continuous growth at the rough interface and the lateral extension growth at the smooth interface, respectively [23]. This results in a regional eutectic Si bias and an extremely inhomogeneous

overall organization. As the injection speed increases, the number of pores and coarse primary $\alpha$-Al phases gradually decrease, and their shape changes to a spherical shape while their size decreases, as shown in Figure 7b,c. This is because the increase in injection speed refines the $\alpha$-Al phase grains, resulting in a shorter primary dendrite arm spacing and smaller secondary dendrite spacing, and also refines the eutectic Si phase, leading to a more uniform overall organization. At an injection speed of 5.0 m/s, as shown in Figure 7d, there are fewer primary $\alpha$-Al phases with a spherical-like shape, reduced eutectic Si segregation areas, and a uniform overall organization.

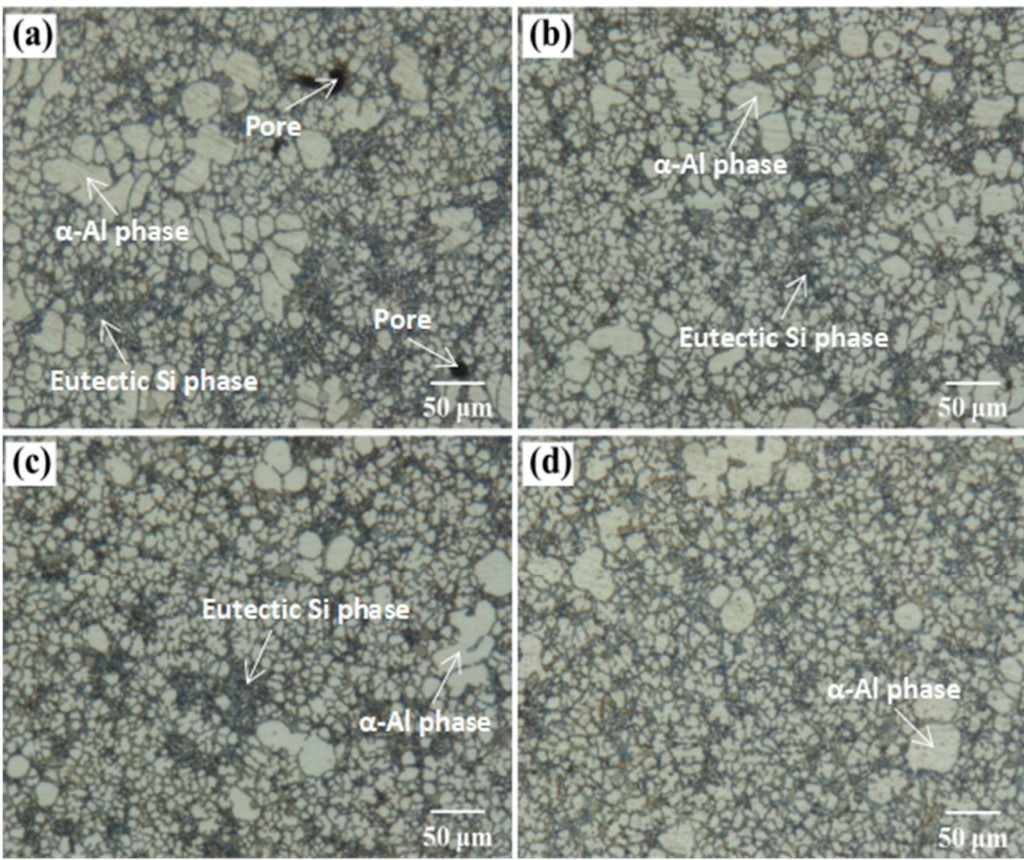

**Figure 7.** Microstructure of thin-walled area at different injection speeds: (**a**) 3.5 m/s, (**b**) 4.0 m/s, (**c**) 4.5 m/s, (**d**) 5.0 m/s.

As shown in Figure 8, the microstructure at different injection speeds in the thick-walled area was observed under a casting temperature of 660 °C, mold preheating temperature of 200 °C, and boost pressure of 850 bar. It can be clearly seen from the figure that the eutectic Si phase at the thick wall is needle-like, similar to the SEM image. Moreover, the average size of grains and the eutectic Si phase in the thick-walled area does not change significantly with the increase in injection speed, but the average diameter of shrinkage pores gradually decreases with increasing injection speed. The average grain size in the thick-walled area is 19.84 μm, 19.82 μm, 20.03 μm and 19.97 μm, the average length of eutectic Si phase is 14.33 μm, 14.52 μm, 14.20 μm and 14.11 μm, and the average diameter of shrinkage pores is 7.97 μm, 7.05 μm, 6.27 μm and 5.79 μm when the injection speed is 3.5 m/s, 4.0 m/s, 4.5 m/s, and 5.0 m/s, respectively, as calculated by the IPP software. Additionally, as the injection speed increases, the number of pores in the thick-walled area first decreases and then increases. When the injection speed is 3.5 m/s, as shown in Figure 8a, the organization is mainly the coarse and massive $\alpha$-Al phase, and the needle-like eutectic Si phase is distributed at the grain boundaries and in the $\alpha$-Al matrix. The overall organization is extremely inhomogeneous, and there are also obvious pores. This is because the solidification sequence of the casting is from the outside to the inside, and when the

injection speed is slow, some areas inside the casting have already solidified, resulting in insufficient subsequent pressurization to compensate for the shrinkage, which, in turn, produces shrinkage pores. As the injection speed increases, it can be seen from Figure 8b,c that shrinkage pores can still be seen in the microstructure at 4.0 m/s. When the injection speed is 4.5 m/s, the shrinkage pores significantly decrease, the $\alpha$-Al phase gradually refines, the number of needle-like eutectic Si decreases, and the number of short-rod or point-like eutectic Si changes. When the injection speed is 5.0 m/s, as shown in Figure 8d, there are fewer coarse $\alpha$-Al phases in the tissue, and the overall tissue is more uniform than at low injection speeds, but there are still shrinkage pores in the tissue.

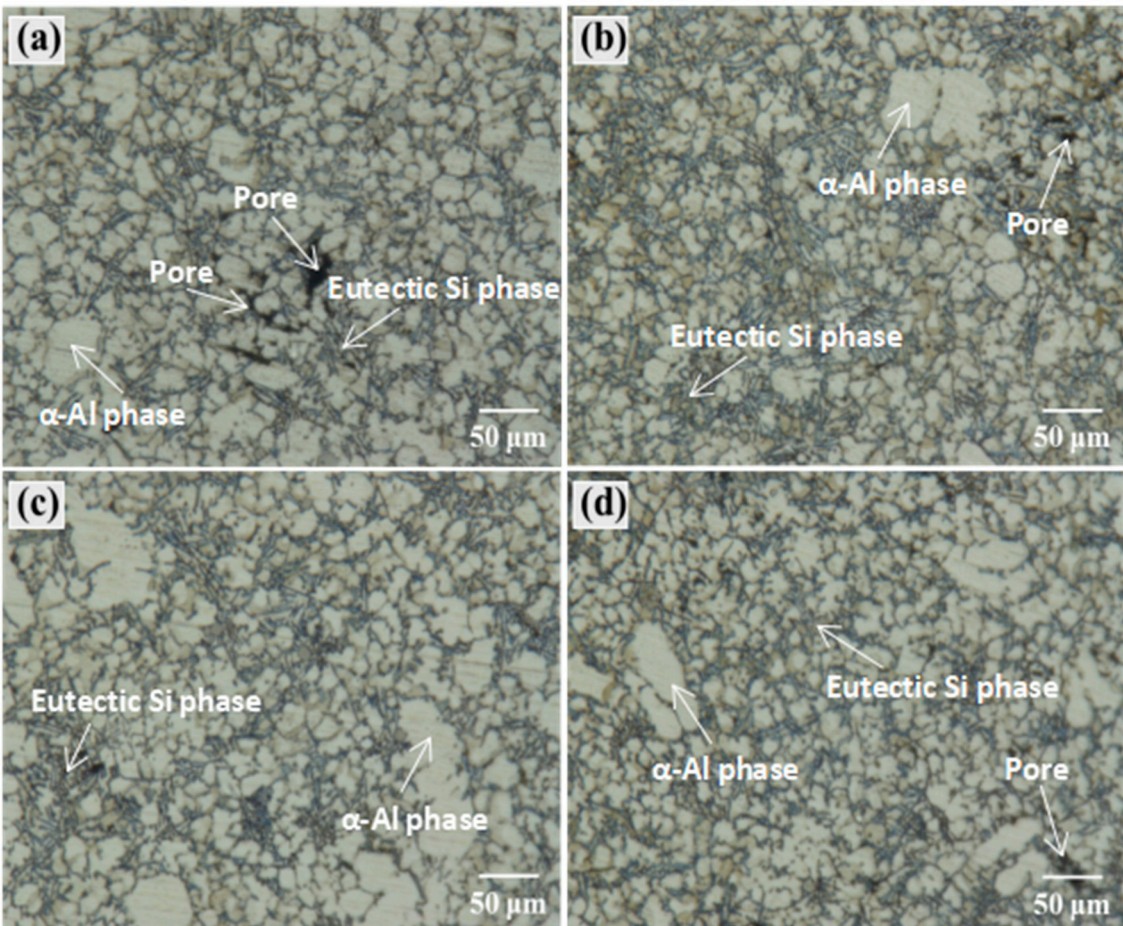

**Figure 8.** Microstructure of thick-walled area at different injection speeds: (**a**) 3.5 m/s, (**b**) 4.0 m/s, (**c**) 4.5 m/s, (**d**) 5.0 m/s.

### 3.2.3. Effect of Boost Pressure on the Microstructure at Different Wall Thicknesses of the Casting

Figure 9 shows the microstructure of the thin-walled area at different boost pressures under a casting temperature of 660 °C, mold preheating temperature of 200 °C, and injection speed of 5.0 m/s. It can be seen from Figure 9 that the average size of grains, eutectic Si phase and shrinkage pores in the thin-walled area decrease gradually with increasing boost pressure. The average grain size in the thin-walled area is 19.01 µm, 18.66 µm, 18.35 µm and 18.13 µm, the average length of eutectic Si phase is 9.94 µm, 9.89 µm, 9.65 µm and 9.43 µm, and the average diameter of shrinkage pores is 7.37 µm, 6.83 µm, 6.64 µm and 5.72 µm when the boost pressure is 700 bar, 750 bar, 800 bar, and 850 bar, respectively, as calculated by the IPP software. Moreover, as the boost pressure increases, the number of pores in the thin-walled area decreases slightly. At a boost pressure of 700 bar, as seen in Figure 9a, large incipient $\alpha$-Al dendrites can be observed in the tissue, the dendrite grain size reaches 167.87 µm, and eutectic Si segregation is also present in the tissue. As the boost pressure increases, this phenomenon persists, as shown in Figure 9b,c, where black areas and coarse

α-Al phases are still visible in the microstructure, resulting in an overall inhomogeneous organization. When the boost pressure is 850 bar, the incipient α-Al phase in the tissue decreases, the shrinkage pores disappear, and the overall tissue is homogeneous.

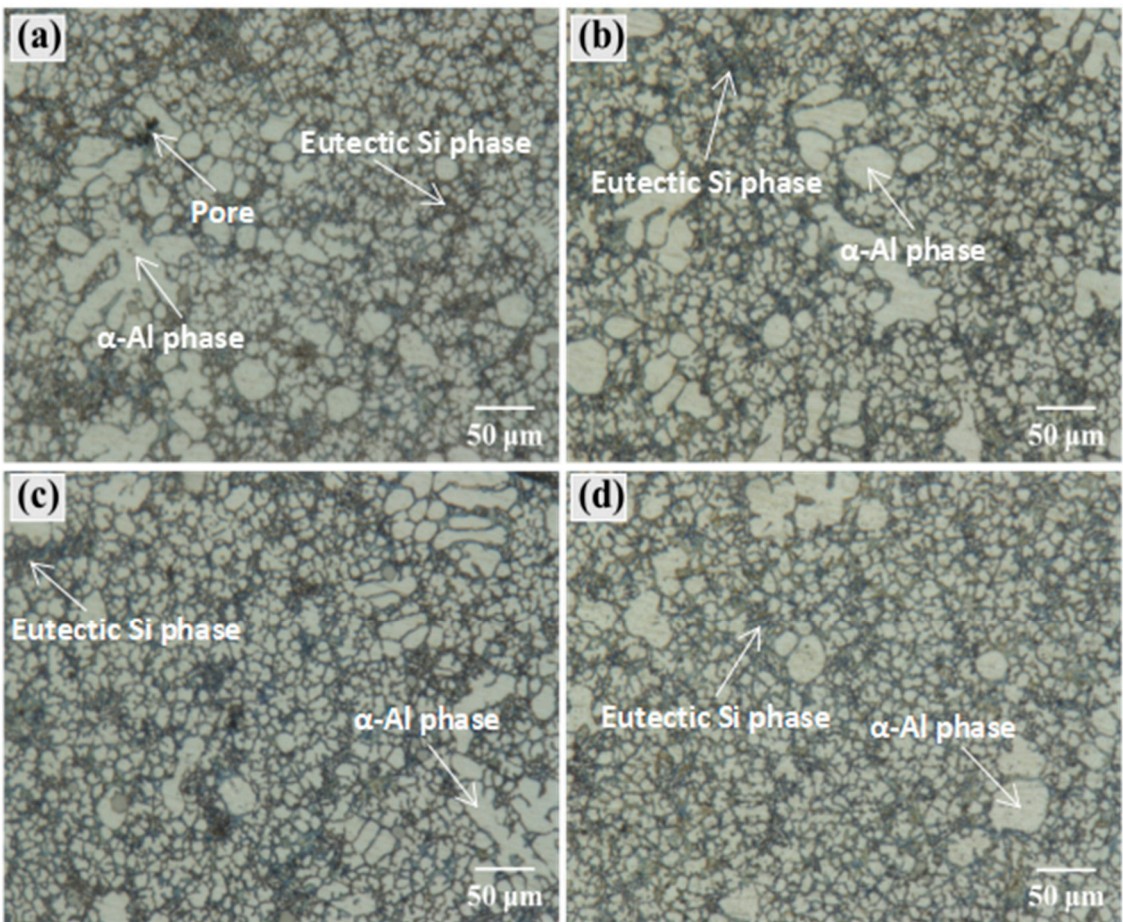

**Figure 9.** Microstructure of thin-walled area at different boost pressures: (**a**) 700 bar, (**b**) 750 bar, (**c**) 800 bar, (**d**) 850 bar.

As shown in Figure 10, the microstructure of the thick-walled area was investigated at different boost pressures under a casting temperature of 660 °C, mold preheating temperature of 200 °C, and injection speed of 5.0 m/s. It can be seen from Figure 10 that the average size of grains and eutectic Si phase in the thick-walled area gradually decrease with the increase in boost pressure, indicating significant grain refinement. The average grain size in the thick-walled area is 20.42 μm, 20.26 μm, 20.05 μm and 19.97 μm, the average length of the eutectic Si phase is 14.36 μm, 14.24 μm, 14.17 μm and 14.11 μm, and the average diameter of shrinkage pores is 6.91 μm, 7.85 μm, 5.97 μm and 5.79 μm when the boost pressure is 700 bar, 750 bar, 800 bar, and 850 bar, respectively, calculated by the IPP software. Additionally, as the boost pressure increases, the number of pores in the thick-walled area increases first and then decreases significantly. At a boost pressure of 700 bar and 750 bar, as shown in Figure 10a,b, large shrinkage pores are observed, and several fine small shrinkage pores are present near the large ones, resulting in large shrinkage porosity in this area. With an increase in boost pressure, the shrinkage porosity decreases and changes from a diffuse to strip-like morphology, as shown in Figure 10c,d, resulting in improved tissue uniformity.

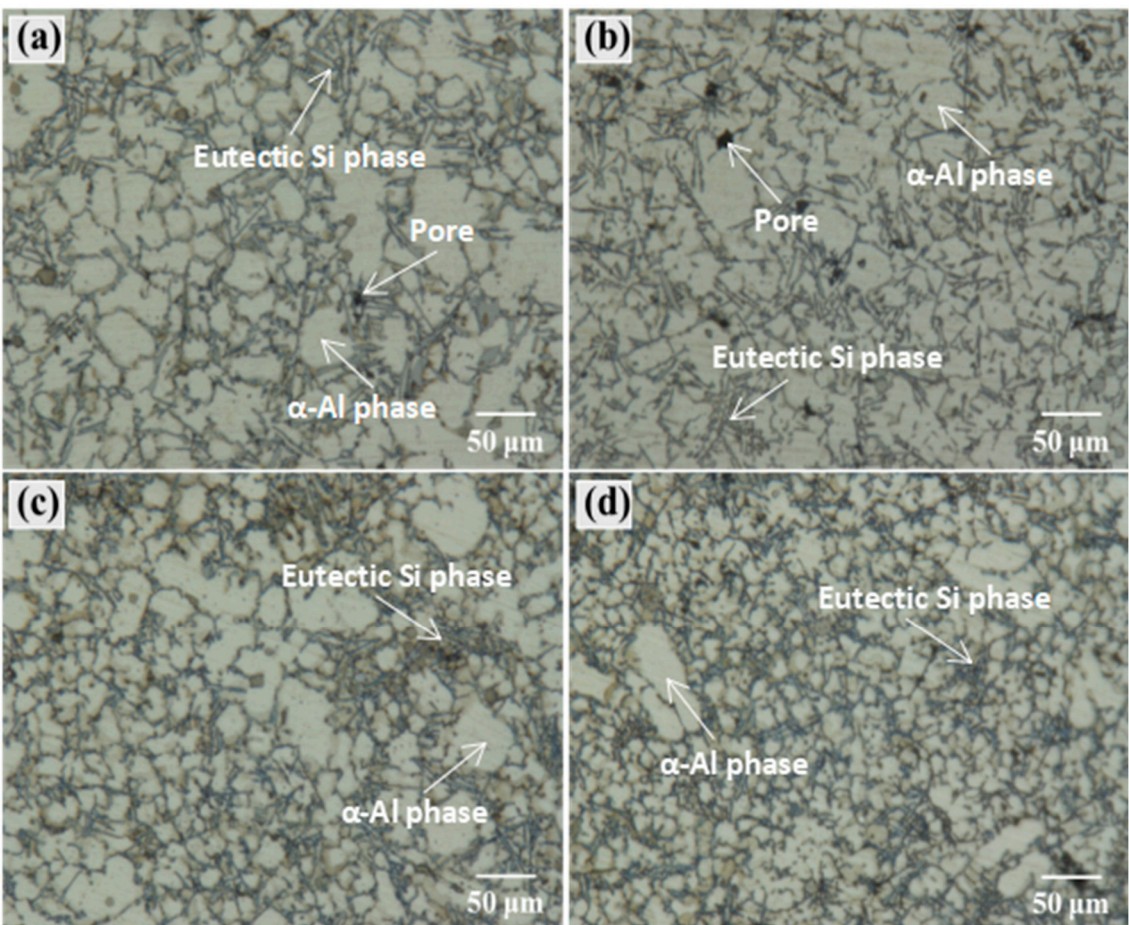

**Figure 10.** Microstructure of thick-walled area at different boost pressures: (**a**) 700 bar, (**b**) 750 bar, (**c**) 800 bar, (**d**) 850 bar.

### 3.2.4. Fracture Morphology Analysis

As shown in Figure 11, the fracture morphology in different areas at injection speeds of 3.5 m/s and 5.0 m/s is presented. By comparing Figure 11a,b, it can be seen that under a low injection speed, the fracture morphology in the thin-walled area of the casting is characterized by cleavage steps, exhibiting brittle fracture. Due to the presence of a large amount of Al-Si and Si-Fe phases in the A380 aluminum alloy, as well as the hard and brittle nature of the long strip-shaped eutectic Si phase, therefore, during tensile loading, significant stress concentrations occur at the sharp ends of the eutectic Si, leading to the formation of cracks. These microcracks grow during tensile loading, and adjacent microcracks connect to form larger cracks [24]; these larger cracks connect with each other, leading to final fracture. When the injection speed is increased to 5.0 m/s, the fracture morphology undergoes a transformation, with a few dimples appearing on the fracture plane. The reason for this phenomenon is that with the increase of the injection speed, the distribution of Al-Si and Si-Fe phases becomes more uniform, and the stress concentration phenomenon at the sharp ends of the eutectic Si is weakened, resulting in it being difficult to fracture. Therefore, a few dimples appear and the elongation of the thin-walled area is improved. By comparing Figure 11c,d, it can be seen that regardless of whether it is a thin-walled or thick-walled area, the fracture morphology is mainly characterized by cleavage steps and a few dimples. Therefore, the difference in elongation between the thick-walled areas at the two injection speeds is not significant.

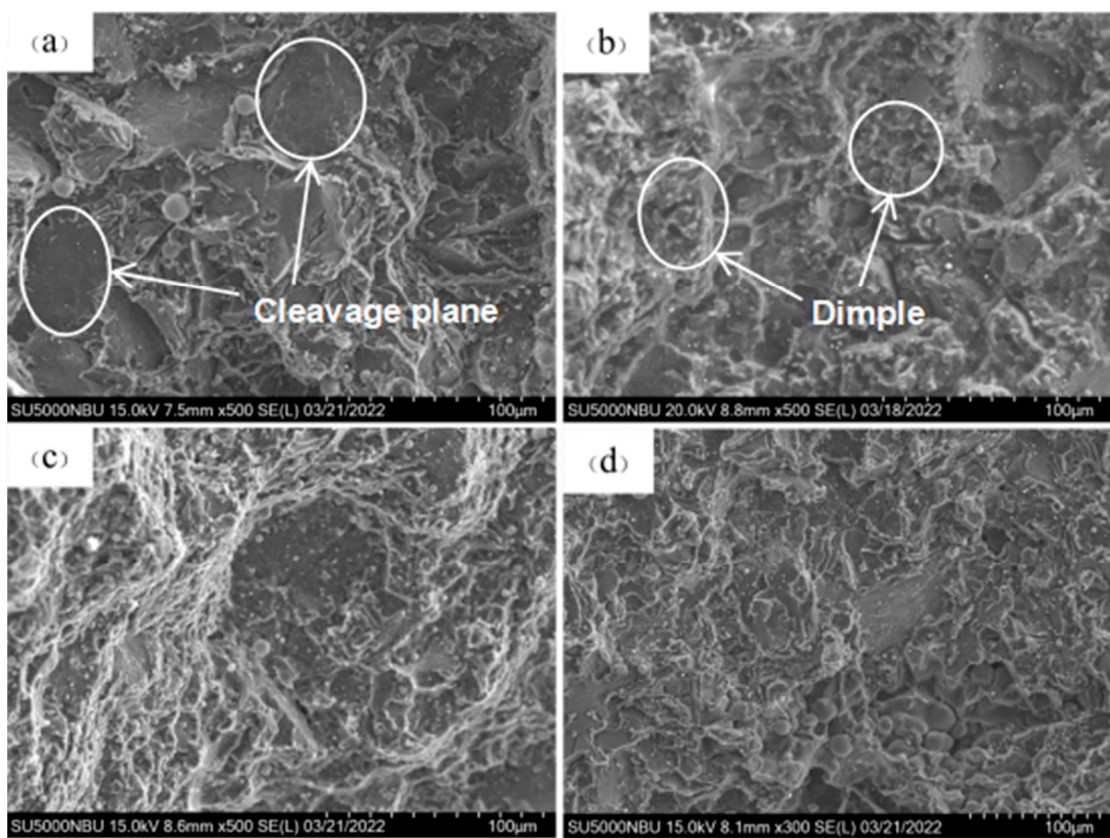

**Figure 11.** Fracture morphologies of different areas at the injection speeds of 3.5 m/s and 5.0 m/s: (**a**) thin-walled area at 3.5 m/s, (**b**) thin-walled area at 5.0 m/s, (**c**) thick-walled area at 3.5 m/s, (**d**) thick-walled area at 5.0 m/s.

As shown in Figure 12, the fracture morphology in different areas under boost pressures of 700 bar and 850 bar is presented. By comparing Figure 12a,c, it can be seen that at 700 bar boost pressure, the fracture morphology in the thin-walled area is characterized by cleavage steps and a few tearing edges, and the fracture mode is a mixture of ductile and brittle fracture with brittle fracture being dominant. This is similar to the fracture morphology of the thin-walled area at an injection speed of 3.5 m/s, so the plasticity is weaker than that at 850 bar. The fracture morphology in the thick-walled area is characterized by cleavage steps and numerous longer tearing edges. The reason for this phenomenon is that the eutectic Si phase in the thick-walled area forms coarse needle-like structures, which are prone to fracture. The tearing edges will extend along the delaminated or fractured eutectic Si phase [25]. The increase in the number of tearing edges reduces the plasticity of the thick-walled area, resulting in slightly poorer plasticity in the thick-walled area at 700 bar compared to the thin-walled area.

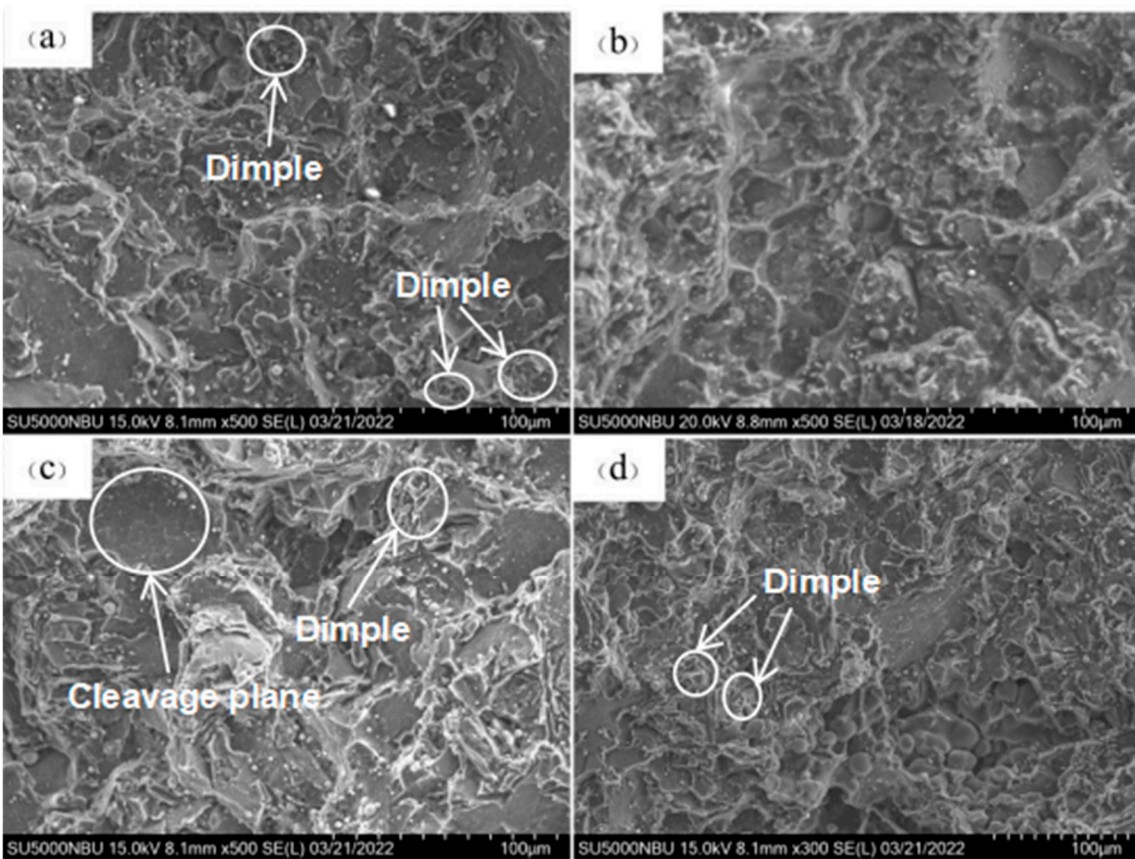

**Figure 12.** Fracture morphologies of different areas at boost pressures of 700 bar and 850 bar: (**a**) thin-walled area under 700 bar, (**b**) thin-walled area under 850 bar, (**c**) thick-walled area under 700 bar, (**d**) thick-walled area under 850 bar.

## 4. Discussion

### 4.1. Effect of Injection Speed on the Mechanical Properties at Different Wall Thicknesses of the Casting

The strength and elongation of the alloy in the thin-walled area gradually increase with the increase of injection speed. The main reason is that at low injection speeds, the alloy contains the Si-Fe phase and the eutectic Si phase is more aggregated, which has a detrimental effect on the strength and elongation of the alloy [26,27]. As shown in Figure 7, with the increase of injection speed the size of the primary α-Al of the alloy gradually decreases, the size of the secondary α-Al also gradually decreases, the degree of aggregation of the eutectic Si phase decreases, the content of Al-Si-Fe-Mn phase gradually decreases, and the strength and elongation of the alloy gradually increase.

In the thick-walled area, the strength and elongation of the alloy increase first and then decrease as the injection speed increases. This is because at low injection speeds, the alloy contains a significant amount of the Al-Si-Fe-Mn phase, and the eutectic Si phase tends to aggregate, negatively affecting the alloy's strength and elongation. As shown in Figure 8, as the injection speed increases, the size of the primary α-Al and secondary α-Al decreases, as does the degree of aggregation of the eutectic Si phase. The content of the Al-Si-Fe-Mn phase also decreases, contributing to the gradual increase in the strength and elongation of the alloy. However, the increase in the number of pores in the alloy as the injection speed increases leads to a decrease in its strength and elongation in the thick-walled area. Taking all these factors into account, the strength and elongation of the alloy in the thick-walled area increase first and then decrease as the injection speed increases.

### 4.2. Effect of Boost Pressure on the Mechanical Properties at Different Wall Thicknesses of the Casting

As the boost pressure increases, the strength and elongation of both the thin-walled and thick-walled areas of the casting gradually increase, with a more significant increase observed in the thick-walled area. Increasing the boost pressure during the extrusion casting process can refine the grain structure and improve the mechanical properties of the alloy. According to the Clausius–Clapeyron equation [28], expressed with Equation (1) as the following:

$$\frac{\mathrm{d}T_\mathrm{P}}{\mathrm{d}P} = \frac{T_\mathrm{m}(V_2 - V_1)}{\Delta H_\mathrm{m}} \tag{1}$$

where $\mathrm{d}T_\mathrm{P}$ is the change in melting temperature under pressure, $\mathrm{d}P$ denotes the change in pressure, $T_\mathrm{m}$ represents the melting temperature at normal pressure, $V_1$ and $V_2$ are the volumes of the corresponding 1 kg solid and liquid phases, respectively, and $\Delta H_\mathrm{m}$ is the latent heat of melting. It can be seen that for alloys with volume shrinkage during solidification, as the boost pressure increases, the melting temperature also increases, resulting in a significant increase in undercooling during solidification of the alloy liquid. The critical nucleus radius $r^0$ during solidification of a metal liquid can be expressed by Equation (2) [29]:

$$r^0 = \frac{2\sigma T_\mathrm{m}}{\rho \Delta H_\mathrm{m} \Delta T} \tag{2}$$

According to reference [28,29], the critical nucleation energy can be expressed by Equation (3):

$$\Delta G^0 = 32\sigma^3 \left[ \frac{2\sigma T_\mathrm{m}}{\rho \Delta T (V_2 - V_1)\mathrm{d}P} \right]^2 \tag{3}$$

where $\rho$ represents the melt density, $\sigma$ denotes the surface tension, $T_\mathrm{m}$ is the melting temperature at normal pressure, $\Delta T$ is the degree of subcooling, and the meanings of the other symbols are the same as in Equation (1). From Equations (2) and (3), it can be seen that increasing undercooling and applying boost pressure can reduce the critical nucleation energy, allowing more atomic clusters to participate in crystal nucleation. At the same time, pressure can decrease the critical nucleus radius, resulting in grain refinement. As the boost pressure increases, the thick-walled area of the casting undergoes greater plastic deformation compared to the thin-walled area, resulting in a more significant refinement of the grain structure. As shown in Figure 10, the Si-Fe phase in the alloy tends to aggregate with large primary $\alpha$-Al and is more difficult to form with fine secondary $\alpha$-Al [30,31]. In addition, the eutectic Si phase tends to form around the fine secondary $\alpha$-Al phase, which is detrimental to the mechanical properties of the alloy. With the gradual decrease in the size of the primary $\alpha$-Al and secondary $\alpha$-Al phases of the alloy, the Al-Si-Fe-Mn phase in the alloy gradually decreases, the eutectic Si phase is gradually distributed evenly, and the mechanical properties of both the thin-walled and thick-walled areas gradually improve.

### 4.3. Effect of Process Parameters on the Microstructure at Different Wall Thicknesses of the Casting

Figures 7 and 8 show that the injection speed has an effect on the microstructure of both thin-walled and thick-walled areas of the alloy. The microstructure at the thin-walled area is significantly affected by the injection speed. As the injection speed increases, the microstructure of the thin-walled area becomes significantly refined, with the size of primary $\alpha$-Al gradually decreasing, the size of secondary $\alpha$-Al also decreasing, and the degree of the eutectic Si phase decreasing. The content of the Al-Si-Fe-Mn phase also gradually decreases. This is mainly due to the shorter filling and solidification time of liquid metal in the thin-walled area of the casting. The large cooling rate of the metal liquid increases the degree of undercooling, increases the number of nucleation cores, improves the nucleation rate, and thus refines the primary $\alpha$-Al, secondary $\alpha$-Al, and eutectic Si. The distribution of needle-like Al-Si-Fe-Mn phase is also more uniform [13,32].

Additionally, as shown in Figures 9 and 10, the boost pressure has a significant effect on the microstructure of both thin-walled and thick-walled areas of the alloy. The

microstructure at the thick-walled area is significantly affected by the boost pressure, and as the boost pressure increases, the microstructure in this area becomes significantly refined [33,34]. The size of primary $\alpha$-Al and secondary $\alpha$-Al gradually decreases, the agglomeration degree of eutectic Si phase decreases, the content of shrinkage porosity gradually decreases, and the average diameter of the shrinkage pores gradually decreases. This is mainly due to the fact that the metal liquid-filled cavity in the thick-walled area of the casting has a longer filling time and cooling time, resulting in a smaller subcooling obtained by the liquid metal, which in turn reduces the number of nucleation cores. Increasing the boost pressure causes a significant plastic deformation in the thick-walled area, resulting in refinement of primary $\alpha$-Al, secondary $\alpha$-Al, and eutectic Si, as well as a more uniform distribution of needle-like Al-Si-Fe phases [35], resulting in a significant improvement in the strength and elongation of the alloy.

## 5. Conclusions

(1) The effects of injection speed and boost pressure on the mechanical properties of castings with different wall thicknesses are significant. Among them, the effect of injection speed on the mechanical properties of the thin-walled area of the casting is more significant. The strength and elongation of the thin-walled area of the casting gradually increase as the injection speed increases. The boost pressure has a greater effect on the mechanical properties of the thick-walled area of the member. The strength and elongation of the thick-walled area of the member increase significantly with the increase in boost pressure.

(2) Increasing the boost pressure can significantly improve the microstructure of the casting, while the boost pressure has a significant effect on the microstructure of the thick-walled area of the casting. The primary $\alpha$-Al phase and secondary $\alpha$-Al phase in the thick-walled area of the casting are gradually refined as the boost pressure increases. In addition, the eutectic Si phase and Al-Si-Fe-Mn phase in the alloy are more uniformly distributed. Therefore, the selection of suitable boost pressure and injection speed has an important influence on the overall forming quality of the casting. In this work, by comprehensively considering the microstructure and overall mechanical properties changes in both thick-walled and thin-walled areas, it is found that the overall properties of the casting are the best when the injection speed is 4.5 m/s and the pressure boosting pressure is 850 bar.

**Author Contributions:** H.L.: Data curation, Writing—review & editing. H.Z.: Writing—original draft, Visualization. W.P.: Conceptualization, Methodology, Investigation, Resources. B.L.: Formal analysis, Supervision. Y.S.: Supervision. L.L.: Writing—review & editing. B.F.: Data curation. Z.Y.: Validation. All authors have read and agreed to the published version of the manuscript.

**Funding:** This project was supported by the National Natural Science Foundation of China (Grants NO. 52075272, 52205386), the Ningbo Science and Technology Plan (Grants No. 2018B10004, 2019B10100), the Provincial Universities Basic Scientific Research Strategy (SJLZ2021002), the Project approved by College Student Science and Technology Innovation Program (SRIP) of Ningbo University in 2022.

**Data Availability Statement:** The data that support the findings of this study are available upon request.

**Acknowledgments:** This work is supported by the Key Laboratory of Impact and Safety Engineering, Ministry of Education and Part Rolling Key Laboratory of Zhejiang, Ningbo University, Ningbo, China.

**Conflicts of Interest:** The authors declare that they have no known competing financial interests or personal relationships that could have appeared to influence the work reported in this paper.

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
