# Peer review of "The Effect of Process Parameters on the Properties and Microstructure of A380 Aluminum Alloy Casting with Different Wall Thicknesses"

_crystals, doi:10.3390/cryst13040587_

Round 1
Reviewer 1 Report
Review of the manuscript "Influence of a process parameter on the properties and microstructure of an A380 aluminum alloy differential housing with different wall thicknesses" by Wenfei Peng, Han Zhang, He Li, Bo Lin, Yiyu Shao, Longfei Lin, Bangji Fu, and Ziming Yu
First of all, I want to note that the authors have done a great and high-quality experimental work. In general, the results are presented well. However, there are some remarks:
1. Replace keywords with more descriptive ones.
2. For parts made of cast aluminum alloys, casting defects have a significant impact on the properties of the metal. It is not clear from the work whether pore size, shrinkage, etc. were evaluated.
3. Need to clarify how many samples were tested to analyze the effect of injection speed on mechanical properties? This spread (for example, relative elongation at a speed of 5 m/s, Fig. 2b) for thick-walled and thin-walled sections can be caused by a casting defect in one of the samples.
4. Line 290 mistake in word "besides".
5. The results of the fructure analysis must be related to the microstructure.
6. Make conclusions on the work more specific, add numerical results.
Author Response
Dear Editor and Reviewer,
We have responded a point-by-point response to your comments, please see the attachment.

Reviewer 2 Report
1. Please comment on the real thickness of the thin-walled area and thick-walled area.
2. Please comment on how many tensile tests were carried out for each diecasting condition.
3. Is the chemical composition identical for all diecast samples?
4. When it comes to the effect of injection speed and boost pressure, fig. 7~9 did not show the size difference of eutectic Si, which might be key to mechanical properties of the samples. Did you ever measure the size of eutectic Si by changing the injection speed and boost pressure?
Author Response

(The authors gave the same response as above.)

Round 2
Reviewer 2 Report
About the author's reply #4, what is the definition of the diameter of eutectic Si? Eutectic Si looks rod-like or needle-like rather than round-shaped as shown in Fig 6. And plus, fig. 6 shows that the thick-walled area has much larger size of eutectic Si than thin-walled area, but the average diameters of eutectic Si measured in the two areas are not that different. Please comment on this matter.
Author Response

(The authors gave the same response as above.)
